# Reproducibility of an Intraoperative Pressure Sensor in Total Knee Replacement

**DOI:** 10.3390/s21227679

**Published:** 2021-11-18

**Authors:** Camdon Fary, Dean McKenzie, Richard de Steiger

**Affiliations:** 1Department of Surgery, Epworth Helathcare, The University of Melbourne, Parkville 3010, Australia; Richard.Desteiger@epworth.org.au; 2Department of Orthopaedics, Western Hospital, Melbourne 3011, Australia; 3Research Development & Governance, Epworth Health Care and Department of Health Sciences and Biostatistics, Swinburne University of Technology, Hawthorn 3122, Australia; Dean.McKenzie@epworth.org.au

**Keywords:** intraoperative pressure sensor, total knee replacement, ligament balancing, soft tissue tension, outcomes

## Abstract

Appropriate soft tissue tension in total knee replacement (TKR) is an important factor for a successful outcome. The purpose of our study was to assess both the reproducibility of a modern intraoperative pressure sensor (IOP) and if a surgeon could unconsciously influence measurement. A consecutive series of 80 TKRs were assessed with an IOP between January 2018 and December 2020. In the first scenario, two blinded sequential measurements in 48 patients were taken; in a second scenario, an initial blinded measurement and a subsequent unblinded measurement in 32 patients were taken while looking at the sensor monitor screen. Reproducibility was assessed by intraclass correlation coefficients (ICCs). In the first scenario, the ICC ranged from 0.83 to 0.90, and in the second scenario it ranged from 0.80 to 0.90. All ICCs were 0.80 or higher, indicating reproducibility using a IOP and that a surgeon may not unconsciously influence the measurement. The use of a modern IOP to measure soft tissue tension in TKRs is a reproducible technique. A surgeon observing the measurements while performing IOP may not significantly influence the result. An IOP gives additional information that the surgeon can use to optimize outcomes in TKR.

## 1. Introduction

For many years, intraoperative pressure (IOP) sensors have been incorporated in total knee replacement (TKR) surgery to measure compartmental tissue tension [1]. The first versions were spring loaded, which the surgeon manually tensioned in fixed positions [2]. Technological advances in microelectronics have made it possible to embed microelectronic sensor arrays within the tibial insert trial that have wireless connectivity to a monitor [3]. Sensors quantify medial and lateral compartment pressures and can define the contact points between the femoral component and tibial insert through the trial joint’s range of motion [4]. This enables the surgeon to adjust soft tissue tension while receiving dynamic visual feedback that is specific to both the TKR design and the patient’s soft tissue envelope.

A modern IOP enables precise quantitative assessment of soft-tissue balance rather than manual (surgeon-defined) assessment [5,6]. Surgeons have been shown to be poor predictors of the true state of balance and over-tensioning of soft tissue is believed to be a significant cause of worse outcomes in TKR [7,8]. Techniques such as kinematic alignment and gap balancing methods are centered around the concept that these methods will optimize knee ligament balance and prevent over-tensioning, which will directly improve patient outcomes [9,10,11]. Poor soft tissue balance may present as either instability of the prosthesis (insufficient tension) or stiffness and pain (over-tension), both of which are recognized causes of revision knee surgery [12]. Insufficient tension leading to instability, subluxation, or even dislocation of the TKR is clinically readily identified intraoperatively compared to over-tension or stiffness. MacDessi et al. [9]. demonstrated that surgeon-defined assessment of tissue tension is a poor predictor of the true soft tissue balance when compared to IOP measurements. However, it is important to note in this paper that despite over 300 consecutive cases using an IOP, there was not a significant learning improvement of manual surgeon-defined knee tension. This could be a reflection of a low reproducibility of the use of an IOP, which was not assessed.

Intraoperatively an IOP informs the surgeon whether soft-tissue corrections, prosthesis manipulation (e.g., downsize, rotation) and/or bone recuts are required to optimize TKR function. These corrections once the trial prosthesis is inserted can be complex and/or challenging, adding to operating time. This could cause the surgeon to unconsciously influence the IOP measurement during dynamic visual feedback while manipulating the trial TKR to decrease operative time. We have examined this by analyzing the correlation between an initial blinded and subsequent unblinded assessment where the surgeon is dynamically visualizing the IOP measurements in real time. Previously published papers using an IOP in the Literature [9,10,11] are with the surgeon making visual assessments in real time and we will be able to validate the reproducibility of this variation in technique to a blinded measurement.

Conventional assessment of soft tissue balance in TKR is traditionally conducted by the surgeon intraoperatively by applying manual tensioning of ligaments throughout a range of motion with the trial implants inserted. This relies on subjective evaluation of tissue tension and is dependent on multiple factors: the magnitude and direction of forces applied, tactile or visual appreciation of laxities, and the ability to control femoral stability/hold the thigh while flexing the knee and holding the tibia and applying a varus or valgus force at the same time. This subjective evaluation is most simple at 10 degrees medial and most challenging at 90 degrees lateral (particularly with the medial parapatellar approach). Another important consideration of manual techniques is that the compartmental tissue tension is significantly altered as the extensor mechanism is not closed or approximated. Surgeons routinely do not close the extensor retinaculum as visual assessment of the trial would not be possible and the tactile assessment becomes more challenging [13]. The use of an IOP eliminates these factors and provides a standardized technique. When the IOP is inserted, the extensor mechanism is approximated. The femur is stabilized and the TKR is moved through its range of motion without varus and valgus forces and the medial and lateral tissue tension is measured separately at 10°, 45°, and 90° flexion.

Studies of modern IOPs to improve outcomes in TKR have shown promising early results. Chow et al. compared sensor-assisted TKR balancing to manual balancing and reported significantly greater improvements in patient-reported outcome scores in the sensor-assisted group [14]. Gustke et al., in a multicenter study of sensor-assisted TKRs, reported significantly better Knee Society Scores and Western Ontario and McMaster Universities Osteoarthritis Index scores than those of the unbalanced knees at one year. They concluded that of all the confounding variables, a balanced TKR was the most significant factor in improving postoperative outcomes [15].

With the increased awareness of the importance of soft-tissue balancing in TKR outcomes, several techniques have evolved and been well documented to manage balance prior to the development of a modern IOP. The gap balancing technique allows for the adjustment of the implant position by force distraction of the collateral ligaments but is not a quantitative evaluation of compartmental pressure [16]. Measured resection is the other technique where bony cuts are based on anatomic references and then the trial implants are used to balance soft tissue tension. Neither technique when trialed accounts for the forces produced by the extensor mechanism when it is closed [3,17,18].

The aims of this study were to assess the intraclass coefficients (ICCs) and reproducibility using an intraoperative pressure sensor in TKR and whether a surgeon could unconsciously influence the measurement. To date there has been no evidence of the reproducibility of the use of a modern IOP in a clinical setting and whether the surgeon could unconsciously influence measurement.

## 2. Materials and Methods

### 2.1. Surgery

A consecutive series of patients undergoing primary unilateral TKA at three hospitals were included in this study. An intraoperative load sensor (VERASENSE, OrthoSensor Inc., Dania Beach, FL, USA) was used in all cases. Exclusion criteria were a tibial plate size 2 or less as the IOP smallest size was 3 (three patients) or if a constrained implant was preoperatively planned. Approval to conduct this study was sought from our local ethics committee.

A total of 76 patients undergoing 80 TKA procedures were assessed. All data was recorded prospectively at the time of surgery. Surgery was performed by fellowship-trained knee arthroplasty surgeons who had been in specialist practice for over 12 years.

Surgeons used their preferred prosthesis and the same technique throughout the study. The aim was to restore a neutral mechanical axis of the lower limb. Once the trial polyethylene spacer was inserted into the trial femoral and tibial components, the surgeons assessed soft tissue balance with their normal technique. The polyethylene spacer would be changed to the thickness that the surgeon defined as a balanced TKR. Then, an IOP insert of the same thickness and shape of the polyethylene was exchanged. Medial retinacular closure was simulated using two clips at the superior and inferior patellar edges to prevent the medial parapatellar arthrotomy from altering intercompartmental loads [13]. Data were recorded at 10°, 45°, and 90° flexion angles by stabilizing the femur and thigh with one hand and moving the tibia and ankle with the other hand, flexing the trial TKR while avoiding varus, valgus, or rotational forces. In Scenario A the same surgeon recorded two blinded measurements. In Scenario B a different surgeon took an initial blinded measurement followed by an unblinded measurement observing the measurements in real time while flexing the TKR.

### 2.2. Data and Statistics

During surgery, data were recorded from the IOP monitor. This involved the intra-articular medial and lateral tissue tensions at 10°, 45°, and 90° flexion as described in Gordon et al. [19]. The data were prospectively collected in a database and subsequently analyzed using Bland–Altman plots as graphic methods for ascertaining agreement between two raters/measures [20]. Intraclass correlation coefficients (ICCs) [21,22] a measure of agreement recently applied in similar orthopaedic studies [23,24,25], were also calculated, using a two-way random effects model, measuring absolute agreement. Each procedure or target was measured by the same rater using two separate blinded measurements (Scenario A) or a blinded and a unblinded measurements (Scenario B). These two sets of ratings were considered to be representative of a larger set of ratings.

Simple comparison of 95% confidence intervals (CIs) for each ICC can be misleading [26], as confidence intervals may overlap yet still be statistically different from each other. Therefore, it is important to measure the 95% CI of the difference between the two ICCs. We used a method first described by Cohen [27]. This method was extended by McKenzie et al. [28] based upon the comparison of correlated or dependent kappa coefficients of agreement for categorical data, obtained from the same sample of patients that was employed in the present study to compare the ICCs within each scenario separately.

The technique is based upon the bootstrap resampling technique of Efron and Tibshirani [29], and our scenarios required drawing 10,000 random sets of two pairs (i.e., first pair: 10 degrees medial blinded 1, 10 degrees medial blinded 2; second pair: 90 degrees lateral blinded 1, 90 degrees lateral blinded 2) from the data. The samples were drawn with replacement; thus, the first bootstrap sample or “resample”, might comprise three copies of patient 1, one of patient 2, none of patient 3 or patient 4, and so on, totaling 48 patients for Scenario A (two blinded) and 32 patients for Scenario B (blinded and unblinded) for each resample. For each resample, the ICCs between patient one at 10 degrees medial blinded 1 and 10 degrees medial blinded 2 (or blinded and unblinded in the case of Scenario B) and patient two at 90 degrees lateral blinded 1 and 90 degrees lateral blinded 2 (ditto) were generated and the difference between them was recorded. A 95% confidence interval was then obtained using the 2.5 and 97.5 percentiles of the 10,000 bootstrap replications (90% CI = 5th and 95th percentiles). This technique has been employed in the analysis of dependent kappas in orthopaedic [30] and psychiatric studies, as well as in the comparison of other types of correlation coefficients for quality control and improvement [31]

Bland–Altman plots, associated summary statistics, ICCs, and comparisons between them were obtained using inbuilt, and in the case of comparison between ICCs, custom written (based on McFarlane et al. [32]) routines within Stata, version 16 (Stata Corporation, College Station, TX, USA, 2019).

## 3. Results

### 3.1. Scenario A—Reproducibility of an IOP

#### 3.1.1. Ten Degrees of Flexion Lateral Soft Tissue Tension

In the Bland–Altman plot (Figure 1), for 10 degree lateral blinded 1 versus blinded 2, the *ideal* mean of the differences (represented by the blue line) between blinded 1 and blinded 2 would be zero, i.e., the two sets of scores would be identical. The *actual* mean of the differences between blinded 1 and blinded 2 is represented by the purple line, and in this case was 0.40, with a standard deviation of 4.20.

The mean of the differences between blinded 1 and blinded 2 was positive, indicating that the reported pressures tended to be higher for blinded 1 than for blinded 2.

The 95% limits of agreement, based upon the mean and standard deviation of the differences, were −7.84 to 8.63. Three cases had differences between blinded 1 and blinded 2 that were higher than the upper 95% limit of agreement.

#### 3.1.2. Ten Degrees of Flexion Medial Soft Tissue Tension

The actual mean of the differences between blinded 1 and blinded 2 is represented by the purple line in Figure 2, and in this case was −2.04 with a standard deviation of 4.09.

The mean of the differences between blinded 1 and blinded 2 was negative, indicating that the reported pressures tended to be lower for blinded 1 than for blinded 2.

The 95% limits of agreement were −10.06 to 5.97. One case had a difference between blinded 1 and blinded 2 that was higher than the upper 95% limit of agreement.

See Appendix A for data and analysis at 45 and 90 degrees of flexion for Scenario A.

The Bland–Altman differences (Table 1) between blinded 1 and blinded 2 tended to be negative (blinded 1 reported pressures lower than those for blinded 2), except for 10 degrees lateral and 45 degrees lateral. Ten degrees medial and 45 degrees medial had the highest (negative) differences of −2.04 and −2.27, respectively.

All ICCs (Table 2) were 0.83 or higher. The coefficient for 90 degrees medial was equal to 0.90, and none were higher.

### 3.2. Scenario B—Unconscious Influence of Surgeons Using IOPs

#### 3.2.1. Ten Degrees of Flexion Lateral Soft Tissue Tension

In the Bland–Altman plot for 10 degree lateral (Figure 3), blinded versus unblinded, the *ideal* mean of the differences (represented by the blue line) between blinded and unblinded would be zero, i.e., the two sets of scores would be identical. The actual mean of the differences between blinded and unblinded is represented by the purple line, and in this case was −1.44 with a standard deviation of 6.46.

The mean of the differences between blinded and unblinded was negative, indicating that the pressures tended to be higher for unblinded than for blinded.

Similar in principle to “process control charts” used in quality control and improvement (e.g., van Schie et al. [33] ), Bland–Altman plots include 95% limits of agreement, represented by the red lower line and the red upper line.

One case had a difference between blinded 1 and blinded 2 that was lower and one case that was higher than the upper 95% limit of agreement.

#### 3.2.2. Ten Degrees of Flexion Medial Soft Tissue Tension

In the Bland–Altman plot for 10 degree lateral (Figure 4), blinded versus unblinded, the actual mean of blinded versus unblinded differences was −1.38 with a standard deviation of 5.52.

The 95% limits of agreement were −12.19 to 9.44.

One case had a difference between blinded 1 and blinded 2 that was lower than the lower 95% limit of agreement and one case that was higher than the upper 95% limit of agreement.

See Appendix B for data and analysis at 45° and 90° flexion of Scenario 2.

All the Bland–Altman means of differences (Table 3) were negative, indicating that reported pressures for blinded were lower than those reported for unblinded, with the exception of 90 degrees medial (0.34).

The largest mean difference between blinded and unblinded was for 90 degrees lateral at −2.19 with a standard deviation of 5.86.

The largest standard deviation (high variation) was for 90 degrees medial, although it had a low mean of 0.34, with a standard deviation of 6.99.

All ICCs (Table 4) were 0.80 or higher. Two coefficients, 45 degrees medial and 90 degrees medial, were equal to 0.90, and none were higher.

### 3.3. Comparison of 10 Degrees Medial vs. 90 Degrees Lateral ICC in Both Scenarios

In both scenarios, the bootstrap 95% confidence interval of the difference between the two ICCs at 10 degrees medial versus 90 degrees lateral contained a value of zero (no difference) and so the difference between the two ICCs was not statistically significant at the 0.05 level.

## 4. Discussion

We have reported for the first time in the literature that the use of a modern IOP (VERASENSE, OrthoSensor Inc., Dania Beach, FL, USA) is a reproducible intraoperative technique to quantify the soft tissue tension of a total knee replacement.

An ICC of 0.8 or greater is the minimum for human reproducibility and an ICC of 0.90 or greater is the minimum for mechanical reproducibility. In both our scenarios, the range of ICCs fell within 0.80 to 0.90, the expected range for human reproducibility. However, the ICCs when two blinded measurements were taken were greater (0.83 to 0.90) than in Scenario 2 (0.80 to 0.90) when the blinded and unblinded measurement were compared. This suggests that IOP reproducibility is improved when measurements are blind.

It is important to know that the use of an IOP is reproducible, particularly as, when an IOP was used in over 300 consecutive cases, there was not a significant learning improvement of manual surgeon-defined knee balance [9]. This could have been caused by poor IOP reproducibility rather than the manual surgeons’ specific techniques or both. A lack of reproducibility of IOP measurements would decrease the surgeon’s ability to improve tactile assessment with future cases.

Surgeon-defined manual evaluation of soft tissue pressure is most simple at 10 degrees medial with direct vision of the medial trial TKR and most challenging at 90 degrees lateral with a standard medial parapatellar approach where the lateral trial TKR is not visible. As expected, we found that in both scenarios, when the two ICCs of these measurements were compared, that there was no statistically significant difference for an IOP when the extensor retinaculum was closed.

An IOP is appropriate in research to quantify balance with new TKR designs or surgical techniques such as medial pivot TKR, kinematic alignment, or robotic-assisted TKR. By combining IOP and robotic-assisted TKR, the surgeon is able to quantify soft tissue balance, alignment, and range of motion of the trial prosthesis intraoperatively. This gives rise to the concept of using multiple technologies for optimal “functional alignment” rather than commitment to a particular concept such as mechanical or kinematic alignment or gap balancing. Gordon et al. reported that robotic-assisted TKR using a gap balancing technique for the initial trial was only balanced in 65% of cases. With subsequent soft tissue corrections and bone recuts, this changed to 87% of cases becoming balanced through the range of motion. However, this resulted in a wide range of coronal alignment which ranged from 6° valgus to 9° varus and the concept of “functional alignment”. Bardou-Jacquet et al. similarly found that, by combining these two technologies in a series of 29 patients, 27 (93%) showed a well-balanced knee in extension at the end of the procedure and 23 (79%) showed a well-balanced knee in terms of flexion and extension with correcting bone cuts alone without soft tissue release.

The aim of the study was to assess the reproducibility of an IOP in two different scenarios. This also allowed us to compare the ICC at 10 degrees medial to 90 degrees lateral, which, respectively, are the most simple and challenging measurements in manual surgeon-defined soft tissue assessment. In both scenarios there was no significant difference, which was expected as the IOP techniques are reproducible.

A limitation of this study was to not have both surgeons independently measuring the IOP of the same patient intraoperatively. Due to the logistics of having two surgeons present for the IOP measurement only and in the interests of decreasing the known risks of serious complications (increased surgical time, number in theatre and scrubbed, etc.), this was not done. The authors believe that, with regards to an operating theatre, there is only one surgeon performing the case and therefore we believed it was important to study what that individual surgeon did rather than compare surgeon to surgeon. It is not possible to examine interrater agreement between the two different scenarios as there was no overlap between the two groups, other than the ICC being slightly greater for Scenario 1, but within human reproducibility for both. The potential for a greater comparison between these two groups is limited as Scenario A compares blinded to blinded measurements and Scenario B compares blinded to unblinded.

## 5. Conclusions

This study has demonstrated that a modern IOP to quantify soft tissue tension in TKR is a reproducible technique. A surgeon recording and observing the measurements in real time while using an IOP may not significantly influence the result. An IOP gives additional information that the surgeon can use to optimize outcomes in TKR.

## Figures and Tables

**Figure 1 sensors-21-07679-f001:**
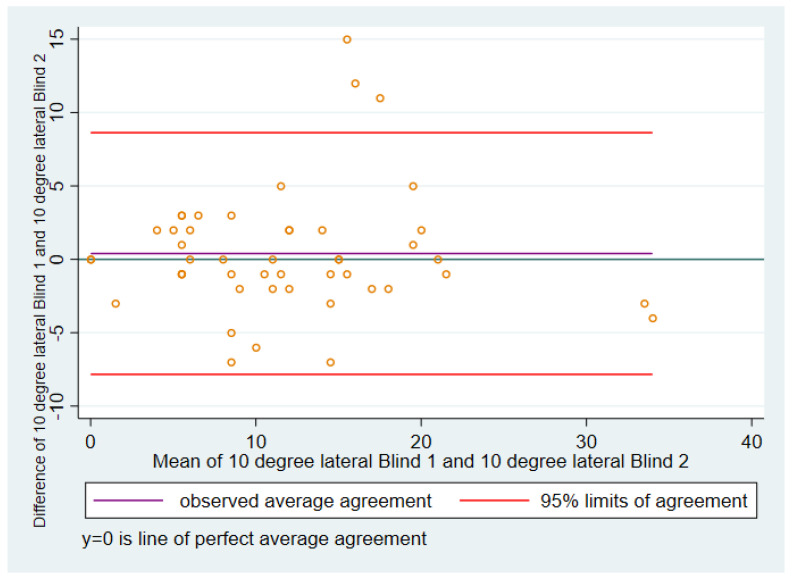
Ten degrees, lateral.

**Figure 2 sensors-21-07679-f002:**
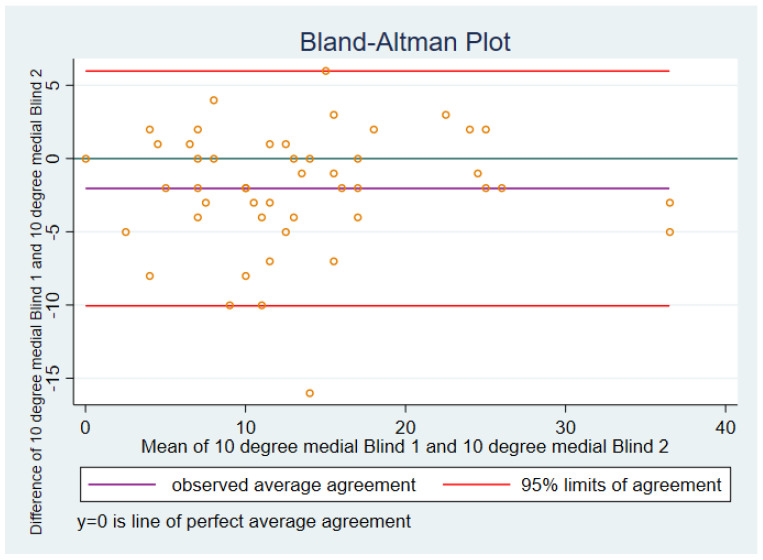
Ten degrees, medial.

**Figure 3 sensors-21-07679-f003:**
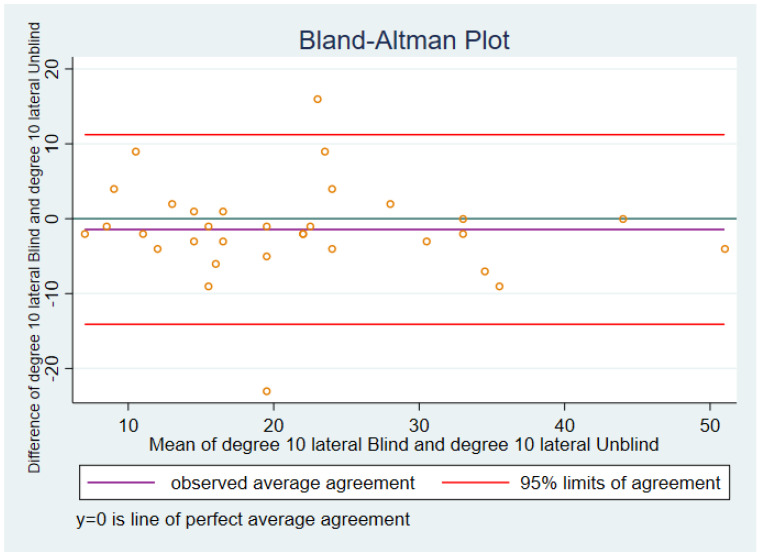
Ten degrees, lateral.

**Figure 4 sensors-21-07679-f004:**
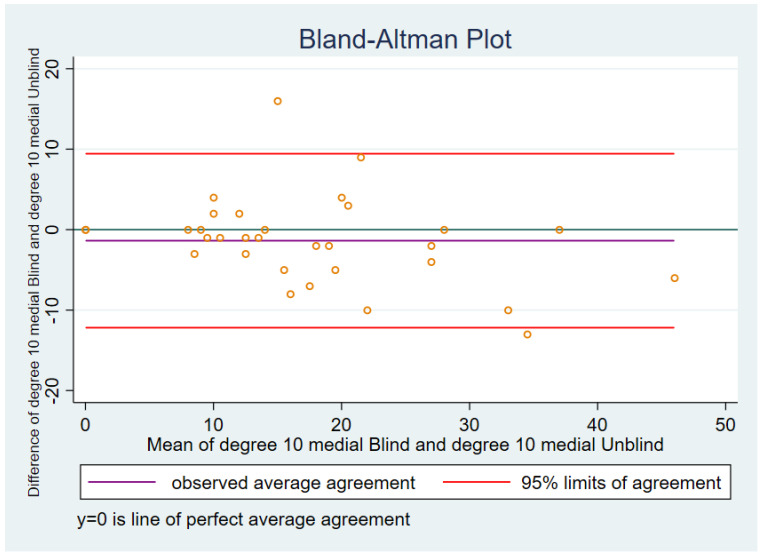
Ten degrees, medial.

**Table 1 sensors-21-07679-t001:** Bland–Altman summary statistics for Scenario A.

Degrees	Side	Mean Difference between Blinded 1 and Blinded 2 (Ideal = 0)	Standard Deviation (SD)	95% Limits of Agreement
10	Lateral	0.40	4.20	−7.84–8.63
10	Medial	−2.04	4.09	−10.06–5.97
45	Lateral	0.25	4.40	−8.38–8.88
45	Medial	−2.27	3.67	−9.46–4.91
90	Lateral	−1.15	5.02	−10.99–8.70
90	Medial	−1.35	4.67	−10.52–7.80

**Table 2 sensors-21-07679-t002:** Intraclass correlation coefficients for Scenario A.

Degrees	Side	ICC	95% CI
10	Lateral	0.84	0.74–0.91
10	Medial	0.85	0.69–0.92
45	Lateral	0.89	0.81–0.94
45	Medial	0.84	0.62–0.93
90	Lateral	0.83	0.71–0.90
90	Medial	0.90	0.82–0.94

**Table 3 sensors-21-07679-t003:** Bland–Altman summary for Scenario B.

Degrees	Side	Mean Difference between Blinded 1 and Blinded 2 (Ideal = 0)	Standard Deviation (SD)	95% Limits of Agreement
10	lateral	−1.44	6.46	−14.1–11.23
10	medial	−1.38	5.52	−12.19–9.44
45	lateral	−1.16	5.66	−12.24–9.93
45	medial	−0.38	5.72	−11.59–10.84
90	lateral	−2.19	5.86	−13.66–9.29
90	medial	0.34	6.99	−13.25–14.04

**Table 4 sensors-21-07679-t004:** Intraclass correlation coefficients for Scenario B.

Degrees	Side	ICC	95% CI
10	Lateral	0.82	0.67–0.91
10	Medial	0.86	0.74–0.93
45	Lateral	0.84	0.70–0.92
45	Medial	0.90	0.81–0.95
90	Lateral	0.80	0.62–0.90
90	Medial	0.90	0.81–0.95

## Data Availability

Restrictions apply to the availability of the data. Data is available from the authors with permission from the Research Development and Governance Unit of Epworth HealthCare.

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
