# Peer review of "Reproducibility of an Intraoperative Pressure Sensor in Total Knee Replacement"

_sensors, 2021, doi:10.3390/s21227679_

Round 1

Reviewer 1 Report

The present study assesses the intra-class coefficients (ICC) and reproducibility with an intraoperative pressure sensor on the TKR, suggesting that the surgeon may be unconsciously influencing the measurement. While the topic is interesting, I believe this work is not sufficient for publication in Sensors for the following reasons:
1. The scientific field of this research/manuscript does not apply to the scope of the Sensors journal.
2. This is a study report that was not independent (it only included measurements from the same surgeons) so it is extremely careful to interpret the results.
3. There are no criteria for extracting the available data, which may result in an inadequate comparison.
4. As the authors say: “This study showed that modern IOP for quantifying soft tissue tension in TKR is a reproducible technique. A surgeon recording and observing real-time measurements with the IOP may not significantly affect the result. The IOP provides additional information that the surgeon can use to optimize the TKR results”. But is the clinical use of this procedure feasible for all cases/patients and will it achieve better treatment outcomes than without an IOP?
5. I propose a few amendments:
- before using the abbreviation for the first time, please specify meaning e.g. IOP in Abstract;
- blinded 1 and blinded 2 are typed differently in text sometimes with small letters sometimes with capital letters;
- the descriptions of figures please place below ones. 

Reviewer 2 Report

We certainly find the topic of interest and we found your work well done and structured.
We also find that your study provide sufficient novel information to merit publication in sensors.

Author Response

Thank you for your comments, interest and suitability for our research article in the Sensor Journal.

Reviewer 3 Report

The purpose of the study is to assess both the reproducibility of a modern IOP sensor and if a surgeon could unconsciously influence measurement. For this reason Consecutive series of TKR were assessed with an IOP in two different scenario: first two blinded sequential measurements in 48 patients and then an initial blinded and then  
unblinded measurement while looking at the sensor monitor screen in 32 patients. Reproducibility was assessed by intraclass correlation coefficients (ICC).

A modern IOP which enables precise quantitative assessment of soft-tissue balance rather is assessed. Outcomes of this study in terms of reliability and repeatability of this measure of soft-tissue balance would impact on the success of the TKR, given that poor soft tissue balance may present as either instability of the prosthesis (insufficient tension) or stiffness and pain (over-tension), that are recognized causes of revision knee surgery. Authors conclude that this study has demonstrated the reliability of a modern IOP to quantify soft tissue tension in TKR. Therefore a surgeon recording and observing the measurements  
in real time while using an IOP may not significantly influence the result. An IOP gives additional information that the surgeon can use to optimise outcomes in TKR.

The topic worth investigating, and results of this may positively impact on clinical practice.

Specific comments

The paper is well written, just minor spell checking is required.

The methods have been clearly reported. The statistics has been correctly applied.

Results were clearly reported. Discussion and conclusion were supported by the results.

Author Response

Thank you for reviewing our research article and suitability for publication. Also, that this TKR soft tissue sensor may be used to positively impact in clinical practice now that the technique has been demonstrated to be reproducible.

We have further spell checked and corrected figures and abbreviations.

Round 2

Reviewer 1 Report

Thanks to the authors for the comments and explanations contained in the cover letter.
I have checked the revised version of the manuscript, and I think the manuscript has been corrected enough to warrant publication in Sensors.